


# Sensitivity Analysis of a Coupled Hydrodynamic-Vegetation Model Using the Effectively Subsampled Quadratures Method

Tarandeep S. Kalra[1], Alfredo Aretxabaleta[1], Pranay Seshadri[2], Neil K. Ganju[1], and Alexis Beudin[3]

[1] U.S. Geological Survey, Woods Hole, MA 02543, USA
[2] University of Cambridge, Cambridge, England, CB2 1PZ, United Kingdom
[3] DEAL Guadeloupe, 97102 Basse-Terre, France

*Correspondence to*: Tarandeep S. Kalra (tkalra@usgs.gov)

**Abstract.** Coastal hydrodynamics can be greatly affected by the presence of submerged aquatic vegetation. The effect of vegetation has been incorporated into the Coupled-Ocean-Atmosphere-Wave-Sediment Transport (COAWST) Modeling System. The vegetation implementation includes the plant-induced three-dimensional drag, in-canopy wave-induced streaming, and the production of turbulent kinetic energy by the presence of vegetation. In this study, we evaluate the

sensitivity of the flow and wave dynamics to vegetation parameters using Sobol' indices and a least squares polynomial approach referred to as Effective Quadratures method. This method reduces the number of simulations needed for evaluating Sobol' indices and provides a robust, practical, and efficient approach for the parameter sensitivity analysis. The evaluation of Sobol' indices shows that kinetic energy, turbulent kinetic energy, and water level changes are affected by plant density, height, and to a certain degree, diameter. Wave dissipation is mostly dependent on the variation in plant density. Performing

sensitivity analyses for the vegetation module in COAWST provides guidance for future observational and modeling work to optimize efforts and reduce exploration of parameter space.

## 1 Introduction

The presence of aquatic vegetation (e.g., mangroves, salt marshes, and seagrass meadows) provides several ecological benefits including nutrient cycling, habitat provision, and sediment stabilization (Costanza et al., 1997). Vegetation provides

habitat for many species of epiphytes, invertebrates, and larval and adult fish (Heck et al., 2003). Seagrass meadows reduce sediment resuspension, thereby stabilizing bottom sediment, increasing light penetration, and improving water clarity in a positive feedback loop (Carr et al., 2010). In addition, aquatic vegetation provides coastal protection by absorbing wave energy (Wamsley et al., 2010).



One approach to implement the influence of aquatic vegetation is by increasing the bottom roughness coefficient in 2DH (depth-averaged) models (Ree, 1949; Morin et al., 2000) with form drag and skin friction partitioning for sediment transport applications (Chen et al., 2007; Le Bouteiller and Venditti, 2015). Increased computational power did not improve the modeling of these two distinct but related drag parameters). To overcome the approximations of these 2-D models and

account for 3-D vertical structures, estuary-scale models have implemented both mean and turbulent flow impacts of vegetation (Temmerman et al., 2005; Kombiadou et al. 2014; Lapetina and Sheng, 2015). Other than exerting drag in the flow-field, the presence of vegetation also results in wave attenuation. This has been studied through a bed roughness approach by simulating wave height decay over vegetation (Möller et al., 1999; de Vriend, 2006; Chen et al., 2007). A more physical description of wave attenuation due to vegetation was developed by Dalrymple et. al (1984) using a cylinder

approach (1984) and applied in spectral wave models to match flume experiments (Mendez and Lozada, 2004; Suzuki et al., 2012; Wu, 2014; Bacchi et al., 2014).

Recently, Beudin et al. (2017) implemented the effects of vegetation in a vertically varying water column through momentum extraction and turbulence dissipation and generation using a 3D hydrodynamic model and accounting for wave dissipation due to vegetation in a spectral wave model. The modeling approach was implemented and tested within the open

source COAWST (Coupled-Ocean-Atmospheric-Wave-Sediment Transport) modeling system that couples the hydrodynamic and wave models. The vegetation module was based on modifications to the flow field through the three-dimensional drag, in-canopy wave-induced streaming, production of turbulent kinetic energy in hydrodynamics model (ROMS) along with exchanges from the wave model (SWAN) to account for wave energy dissipation. The equations that describe the influence of vegetation in flow field depend on vegetation properties (plant density, plant height, plant diameter,

and plant thickness) as input from the user.

Based on the physical conditions (seasonal and environmental), these vegetation properties can be highly variable, yet measurement can be time-consuming in realistic settings. Therefore, it is necessary to identify which properties have the greatest influence on the resulting flow dynamics. The current work aims to perform a systematic sensitivity study to quantify the effect of changing the vegetation properties on the resulting hydrodynamic parameters. Sensitivity studies

represent an important and necessary step in the development of coupled hydrodynamic and wave models. They provide information regarding the controlling dynamics in complex systems and highlight the aspects of the model that require further development. The results of the sensitivity analysis can be used to select and rank the most important parameters for calibration. Quantifying the effect of parameter changes in different aspects of the model solutions allows for more effective model implementation and reduces the range of parameters that future simulations require. Two conditions are required for

the model to display a significant sensitivity: (1) a sufficient modification of one of the forcing parameters, (2) a change in the leading terms of the dynamic equations of the model. While modifying the forcing parameters by a sufficient amount is required, the modification should remain within the natural range of variability of the parameters.

Several mathematical techniques have been utilized to perform sensitivity analysis. Bryant (1987) used scaling analysis in an idealized domain and forcing to find that closure parameters such as vertical diffusivity and wind stress curl





were important controlling factors in thermohaline circulation. Bastidas et al. (1999) used multicriteria methods to find sensitivity of parameters in land surface schemes to model a complex surface. The results of sensitivity analysis showed consistency with physical properties for two different field sites and helped identifying insensitive parameters that led to an improvement in model description. Fennel et al. (2001) incorporated adjoint methods to perform sensitivity studies to

improve the model parameters in an ecological model such that it can be applied on a wider range of problems. The results of the optimization process when applied to real measurements at the Atlantic Bermuda site gave poor results in conjunction with error estimation led to the understanding that the underlying model formulation was inadequate. Mourre et al. (2008) performed multiple simulations based on realistic variation of forcing field to calculate the influence of model parameters on sea surface salinity. The metric used to measure the sensitivity was based on RMS difference between the reference and

modified model parameter. The results showed that lateral salt diffusivity had the strongest impact on surface salinity model response. Rosero et al. (2009) investigated the sensitivity of three different versions of a land satellite model (Noah LSM) with the choice of nine different sites based on different conditions (soil, vegetation, and climate). They utilized the Monte Carlo method to generate the first order Sobol' indices to estimate the model sensitivity. The results showed that the optimal parameter values differed between different versions of the models and for different sites along with quantifying the nature

of interactions between parameters.

The choice of sensitivity analysis methodology depends on multiple factors such as the computational cost of running the model, the characteristics of the model (e.g., nonlinearity), the number of input parameters, and/or the potential interactions between parameters. Saltelli et al. (2008) provided a comparison of different sensitivity analysis methodologies and the optimal setup for specific combinations of parameters and model. They described variance-based techniques as

providing the most complete and general pattern of sensitivity for models with a limited number of parameters. Sobol' indices (Sobol', 1993), as a form of variance-based sensitivity analysis, provide a decomposition of the variance of a model into fractions that can be assigned to inputs or combinations of inputs.

One of the challenges associated with a Monte Carlo approach to computing the Sobol' indices is the large number of model evaluations required for approximating the conditional variances. Techniques that involve approximating the global

response of the model with a polynomial, and then using its coefficients to estimate the Sobol' indices have proven to be both efficient (reduced computational cost) and accurate (Sudret, 2008). In this paper, we use a set of least squares polynomial tools based on subsampling to estimate our global polynomial response (Seshadri et al., 2017a). Then, the coefficients of the polynomial are used to compute the Sobol' indices. These tools are implemented in the open source package, Effective Quadratures (Seshadri et al., 2017b).

The paper is organized as follows: the methods are discussed in section 2, including the numerical model, Effective Quadratures, and simulation design. In section 3, we present the results of sensitivity analysis from various simulations and in section 4, we discuss the impact of these results. Finally, in section 5 we summarize the work and outline areas of future research.



## 2 Methods

### 2.1 COAWST implementation of vegetation model parameterization:

Beudin et al. (2017) implemented a hydrodynamic-vegetation routine within the open-source Coupled Ocean-Atmosphere-Wave-Sediment Transport (COAWST) numerical modeling system (Warner et al., 2008). The COAWST framework utilizes ROMS (Regional Ocean Modeling System) for hydrodynamics and SWAN (Simulating WAves Nearshore) for modeling waves coupled via the Model Coupling Toolkit (MCT) generating a single executable program (Warner et al., 2008).

ROMS (Regional Ocean Modeling System) is a three-dimensional, free surface, finite-difference, terrain-following model that solves the Reynolds-Averaged Navier-Stokes equations using the hydrostatic and Boussinesq assumptions (Haidvogel et al., 2008). The transport of turbulent kinetic energy and generic length scale are computed with a generic length scale (GLS) two-equation turbulence model. SWAN (Simulating WAves Nearshore) is a third-generation spectral wave model based on the action balance equation (Booij et al., 1999). The effect of submerged aquatic vegetation in ROMS is to extract momentum, add wave-induced streaming, and generate turbulence dissipation. Similarly, the wave dissipation due to vegetation modifies the sink term of the action balance equation in SWAN. All these sub-grid scale parameterizations account for changes due to vegetation in the water column extending from the bottom layer to the height of the vegetation. SWAN only accounts for wave dissipation due to vegetation at the bottom layer. The parameterizations used to implement the effect of vegetation in both ROMS and SWAN models are mentioned in Table 1 and detailed in Beudin et al. (2017).

The coupling between the two models occurs with an exchange of water level and depth averaged velocities from ROMS to SWAN and wave fields from SWAN to ROMS after a desired number of time steps (Fig. 1). The vegetation properties are separately input in the two models at the beginning of the simulations. Once the simulation begins, the vegetation inputs are sent to SWAN at the end of each coupling cycle.

### 2.2 Method for Sensitivity Analysis: Polynomial Least Squares

Polynomial techniques are ubiquitous in the field of uncertainty quantification and model approximation. They approximate the response of some quantity of interest with respect to various input parameters using a global polynomial. From the coefficients of the polynomial one can then estimate the mean, variance, skewness and higher order statistical moments (see Smith 2014; Geraci et al. 2016). In this paper our interest lies in statistical sensitivity metrics called first order Sobol' indices (Sobol', 1993) that are derived from the conditional variances of the parameters of the model. These indices, which are the same in number as the input parameters to the model; quantitatively rank the input parameters based on their contribution to the model output. Thus, parameters that exhibit higher first order Sobol' index value are more important than those with a lower value. In addition to first order Sobol' indices, second order and third order indices may also be computed, with the understanding that all these indices must sum to unity. However, should the sum of the first order Sobol' indices themselves be close to unity, indicating that the contribution of the higher order Sobol' indices, and thus higher order interaction



between model input parameters; is low and quite possibly negligible. In this paper, the first order indices are computed from a global polynomial model using Effective Quadratures.

There are two attributes to any data-driven polynomial model: the choice of the polynomial basis and the strategy for estimating the coefficients of the polynomial. The bases used in Effective Quadratures are orthogonal polynomials; i.e.

orthogonal with respect to the weight of the input parameter. For example, if one of the input parameters is prescribed with a Gaussian distribution, then a Hermite orthogonal polynomial basis would be used; likewise for a uniform distribution, Legendre polynomials are used. The rationale behind selecting polynomials that are orthogonal with respect to the input weight is that it reduces the number of model evaluations required for estimating statistical moments. Details on the exponential convergence in moments when matching the orthogonal polynomial with its corresponding weight can be found

in Xiu et al. (2002).

The coefficients for the polynomial expansion are typically approximated using an integral over the input parameter space using quadrature rules. When the number of input parameters is greater than one, tensor grid or sparse grid based quadrature rules may be used to approximate these integrals. However, the cost of tensor grids grows exponentially with dimension, i.e., a four-point quadrature rule in three dimensions has $5^3$ points, in four dimensions has $5^4$ points and so on.

While some alleviation can be obtained using sparse grids, in this paper a more efficient sampling technique is used: effectively subsampled quadratures (abbreviated to Effective Quadratures).

The method of Effective Quadratures determines points for approximating the integral by subsampling well-chosen points from a tensor grid, and evaluating the model at those subsamples. These well-chosen points are obtained via a QR column pivoting heuristic (Seshadri et al. 2017a). Once the coefficients are estimated, the Sobol' indices can be readily

computed (Sudret 2008).

### 2.3 Range of input vegetation properties for sensitivity analysis

Prior to performing the simulations for estimating Sobol' indices described above, a range of vegetation inputs that would impact the model response needs to be chosen. Kennish et al. (2011) provided an annual variation of three of the four

vegetation properties (density, height, diameter) based on *Zostera marina* growth in the Barnegat Bay-Little Egg Harbor estuary. The thickness of *Z. Marina* is an order of magnitude lower than its diameter (J. Testa, personal communication). Based on the published data, the range of the vegetation model inputs is as follows:

1.  Density $(n_v)$= [38.2 - 250.4] stems/m$^2$.
2.  Height $(l_v)$ = [0.16 – 0.32] m.

3.  Diameter $(b_v)$ = [1.0 – 2.0] cm.
4.  Thickness $(t_v)$ = [1.0 – 2.0] mm.

For the sensitivity analysis, a combination of these ranges of inputs is chosen to configure different simulations in an idealized test case, described below.



### 2.4 Test case configuration

An idealized rectangular model domain of 10 km by 10 km with a 3 m deep basin is chosen. The grid is 100 by 100 in the horizontal (100 m resolution) and has 60 vertical sigma-layers (uniformly distributed) leading to 0.05 m resolution in the vertical. The vertical resolution of 0.05 m allows a plant height of 0.27 m to be distributed over 6 vertical layers while the

shortest height is restricted to 2 vertical layers. A square patch of vegetation (1 km by 1 km) is placed in the middle of the domain. The ROMS barotropic and baroclinic time steps are respectively *0.05 s and 1 s*, while the SWAN time step and the coupling interval between ROMS and SWAN are 10 min. The friction exerted on the flow by the bed is calculated using the SSW bbl in the bottom boundary layer formulation (Warner et al., 2008). The bottom boundary layer roughness is increased by the presence of waves that produce enhanced drag on the mean flow (Madsen, 1994; Ganju and Sherwood, 2010). The

vegetative drag coefficients ($C_D$) in the flow model and the wave model are set to 1 (typical value for a cylinder at high Reynolds number). The bed roughness is set to $z_0=0.05$ mm, which corresponds to a mixture of silt and sand (Soulsby, 1997). The turbulence model selected is the $k-\varepsilon$ scheme (Rodi, 1984).

The model is forced by oscillating the water level on the northern edge with a tidal amplitude of 0.5 m and a period of 12 h. Waves are also imposed on the northern edge with a height of 0.5 m, directed to the south (zero angle), with a period

of 2 s. The test case setup is simulated for 2 days to obtain a tidally steady state solution. These simulations require 40 CPU hours on Intel Xeon(R) X5650 2.67 GHz processors running on 24 parallel processors. The output parameters, described below, are chosen to display a significant sensitivity to change in the leading terms of the dynamic equations of the model.

### 2.5 Choice of COAWST model response to vegetation inputs

The output parameters used to investigate the vegetation model sensitivity are chosen to reflect the first order effects of vegetation on the hydrodynamics and waves. The presence of vegetation affects the output parameters in different physical ways (Table 1) as:

1. Wave dissipation: The presence of vegetation dissipates wave energy, thereby reducing the wave height, increasing the wavelength, and reducing wave steepness.

2. Kinetic energy: The presence of vegetation generates a drag force. This leads to a decrease in kinetic energy within and behind the vegetation patch.

3. Water level: As the wave energy (and momentum) flux decreases due to bottom friction, the mean water level increases to balance the decrease in wave and kinetic energy. The flow decelerates in front of the patch and in the wake of the patch while it accelerates around the edges of the patch, leading to a water level gradient.

4. Turbulent kinetic energy (TKE): The presence of vegetation leads to a reduced turbulent kinetic energy in front and within the patch. The enhancement of TKE inside the boundary layer is not captured with the current resolution.

The response impact to change in the inputs during each simulation is computed by calculating the percentage difference of model response for each simulation from the minimum value of all the simulations. The model response is obtained in and around the vegetation patch and averaged over the last tidal cycle. The change in model response of water





level is computed by finding the maximum water level difference in and around the vegetation patch. In addition, the variability of model response with given vegetation inputs in different simulations is calculated through standard deviation of model response in and around the vegetation patch over the last tidal cycle. The standard deviation in TKE is depth averaged to provide a 2-D field.

## 3 Results

### 3.1 Setting up simulations with different vegetation inputs

Using the range of input parameters described above (section 2.3), and assuming all the inputs are uniformly distributed over their ranges, a matrix of design of experiment values (Table 2) was determined using Effective Quadratures. A total of 15 simulations were found to be required; as this corresponds to the number of coefficients in a 4D polynomial with a maximum order of 2. In general, the number of coefficients $n$, is given by the formula

$$n = \binom{d + o}{d},$$

where $d$ is the number of dimensions and $o$ is the maximum order (assuming it is isotropic across all dimensions).

### 3.2 Model response from all the simulations

The 15 simulations (parameter choice of each simulation in Table 1) are performed to provide model response from the four chosen output variables. The results for the model response are computed for the last tidal cycle (a total of 3 tidal cycles are required for achieving steady state).

1. Wave dissipation: The percentage change in wave dissipation (method described in Section 2.5) from all simulations relative to the minimum value of wave dissipation varied between 75 and 600% (Fig. 3(a)). Simulation 2, which includes the combination of lowest density ($n_v$ = 62.1 stems/m$^2$) and shortest height (0.174m), incurred the lowest wave dissipation, while simulation 3, which involved the combination of highest density ($n_v$ = 226.5 stems/m$^2$) and tallest height (0.295 m), resulted in highest wave dissipation. The highest amount of variability in wave dissipation occurs in front of the vegetation patch (Fig 4). This is also the region where the highest amount of wave energy is dissipated due to the presence of vegetation patch.

2. Kinetic energy: The percentage change in kinetic energy from all simulations relative to the minimum kinetic energy varied between 5.0 and 34.0 % (Fig. 3b). Simulation 8, performed with an intermediate value of density along with the highest values of height, diameter, and thickness, results in the least amount of kinetic energy (lowest velocities). The highest kinetic energy in and around the vegetation patch is obtained in simulation 2 with a combination of plant inputs ($n_v$ = 62.1 stems/m$^2$, $l_v$ = 0.174 m, $b_v$ = 3 mm and $t_v$ = 0.6 mm). Simulation 2 causes the least amount of extraction of momentum with a combination of lowest plant density, height, and diameter values. The variability in kinetic energy from all the simulations is observed at a cross section along the vegetation patch (Fig. 5). The enhanced variability is present throughout the water column where the vegetation patch exists. Similar to the variability in wave dissipation, the highest amount of

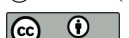



variability occurs in front of the vegetation patch. The region of maximum variability occurs from a depth of (0.25 – 1.25) m.

3. Water level: The percentage change in maximum water level difference from all simulations relative to the minimum of the maximum water level difference varied between 3.0 and 18.0 % (Fig. 3c). The minimum value of model response is obtained from simulation 13 that includes a combination of plant inputs ($n_v$= 62.1 stems/m$^2$ (minimum), $l_v$= 0.174 m (intermediate), diameter=6 mm (minimum), $t_v$= 0.3 mm (minimum). Three simulations that give low values of maximum gradient in water level are simulations 2, 6, and 11. Simulations 2 and 6 both involve the lowest plant density values (i.e. $n_v$= 62.1 stems/m$^2$). On the other hand, simulation 8 ($n_v$=144.3 stems/m$^2$, $l_v$= 0.3 m (maximum), $b_v$=9 mm, $t_v$= 0.9 mm) accounts for the highest value of maximum water level difference. The variability in water level (Fig. 6) is highest around the lobes of the vegetation patch where the water level adjusts owing to a changes in velocity around the patch. Behind the vegetation patch as well, the variability increases as the water level adjusts due to a decrease in values result in the highest variation in water level.

4. Turbulent Kinetic Energy (TKE): The percentage change in TKE from all simulations relative to the minimum TKE varied between 0.5 and 10.0 % (Fig. 3d). Simulation 8 gives the least amount of TKE. These results are similar to the ones above with kinetic energy change where simulation 8 implemented a combination of plant $n_v$= 144.3 stems/m$^2$, $l_v$= 0.3 m, $b_v$= 9 mm and $t_v$= 0.9 mm. The lowest TKE is obtained in and around the vegetation patch in simulation 2 with a combination of plant inputs ($n_v$ = 62.1 stems/m$^2$, $l_v$ = 0.2 m, $b_v$= 3 mm and $t_v$ = 0.6 mm). Simulation 2 causes least amount of dissipation of turbulence with a combination of lowest plant density, height and diameter. The variability in TKE peaks in front of the vegetation patch (Fig. 7) as the different simulations dissipate turbulence in substantially different ways in front of the patch (leading edge phenomenon). The changes in turbulence mixing caused by the presence of the vegetation patch are close to zero inside the patch (all simulations dissipate similar amounts of turbulence).

### 3.3 Quantifying sensitivity using Sobol' Indices

Following the variability in model response from different simulations, the sensitivity to input vegetation parameters can be quantified with the use of first order Sobol' indices that are obtained by taking advantage of the Effective Quadratures approach. Sobol' indices are individually computed for all the model responses. The first order Sobol' indices for all the model responses (Table 3) add up to above 0.9 implying that they account for 90% of the variability in model response for given vegetation property inputs. Because the sum of Sobol' indices is above 0.9, the rest of the variability is caused by second order and third order indices with the understanding that all these indices must sum to unity. The highest model sensitivity (Table 3) is caused by vegetation density ($n_v$) and height ($l_v$). This is indicated by high Sobol' index associated with the two inputs that account for over 80% of the sensitivity to all the model outputs. The vegetation diameter ($b_v$) shows about 12-15 % sensitivity to kinetic energy, water level change, and turbulent kinetic energy. Thickness ($t_v$) showed the least impact on all the chosen model outputs.





## 4 Discussion

### 4.1 Variability in model response from sensitivity analysis

From the different simulations performed during sensitivity analysis, there is a high amount of variability in front of the vegetation patch in wave dissipation, KE, and TKE (Figs. 4, 5 and 7). This is a result of high amount of wave dissipation and

flow deceleration in front of the vegetation patch. The cross-sectional plane of the domain highlights that the variability in KE occurs throughout the water column (Fig. 5); highlighting the 3-D impact of vegetation inputs in the idealized domain. Interestingly, the highest amount of variability in KE (Fig. 5) occurs at depths between $0.4 - 1.3$ m at y = 5.6 km while the maximum vegetation height in all the simulations is 0.3 m. This result indicates that the highest amount of KE variability occurs above the vegetation patch due to different vegetation heights that are involved in different simulations. The

variability in water level change is high around the lobes and behind the vegetation patch (Fig. 6). The variability in these regions is because the change in local flow velocity is adjusted by a change in water level around the vegetation patch.

Overall, the combination of highest vegetation density and vegetation height (Simulation 8) causes the highest wave dissipation, highest TKE, and maximum water level change and subsequently, lowest kinetic energy. Conversely, a combination of lowest vegetation density and vegetation height (Simulation 2) causes the least amount of wave dissipation,

lowest TKE, and maximum water level change. The consistency in these flow patterns observed in the model response highlights the influence of the choice of vegetation inputs. This is further confirmed by observing the variation in velocity profiles with depth (Fig. 8a) at a particular time instance during flood tide. Simulation 8 results in the lowest velocity of 0.06 m/s while simulation 2 results in the peak flood velocity of 0.11 m/s. Consequently, the gradient of velocity with respect to depth is highest at all depths for simulation 8 while lowest for simulation 2 (Fig. 8b). The gradient reaches a maximum value

at the bottom layer.

### 4.2 Understanding vegetation parameterization to interpret Sobol' Indices

The parameterization of vegetation processes shows the relationship of model output and vegetation inputs (Table 1). The parameterizations involving extraction of momentum, turbulence production and turbulence dissipation are directly affected

by vegetation density ($n_v$), width ($b_v$) and thickness ($t_v$). Because these mechanistic processes occur at the blade scale, the dependence on vegetation height ($l_v$) is implicitly included in the parameterizations. Sobol' indices provide quantifiable information and show that vegetation height, density and width (in decreasing order of importance) are pertinent in accurately computing kinetic energy (KE), turbulent kinetic energy (TKE) and water level. An accurate representation of KE and TKE has direct ramifications on estimating sediment transport while water level estimates can affect storm surge

predictions.

The high sensitivity of wave dissipation to vegetation density highlights the need of accurate density representation to attain wave attenuation estimates; especially in open coasts. SWAN computes wave dissipation due to vegetation as a bottom layer effect. Therefore, the height of the vegetation does not seem to affect wave dissipation to the same extent as other model outputs: KE, TKE, and water level. In addition, the equation representing wave dissipation process in SWAN is



independent of vegetation thickness; thus corresponding to the lowest Sobol' index. Vegetation thickness only appears in the turbulence dissipation term (Table 1) while modifying the turbulence length scale and consequently it has the least effect on any of the model response.

**4.3 Linear curve fitting to complement Effective Quadratures based sensitivity analysis**

To complement the results of the Sobol' indices calculation linear fits to the data are conducted. The main parameter contributing to the wave dissipation variability is the shoot density explaining over 80% of the variability ($R^2$=0.81). The least squared fit to wave dissipation that included all parameters combined explains 98% of the variability. When fitting models, it is possible to increase the explained variance by using more complex fitting models adding parameters, but doing so may result in over-fitting. The Bayesian Information Criterion (BIC; Schwarz, 1978; Aretxabaleta and Smith, 2011) provides a non-subjective metric for the best fit by penalizing over-parameterization. BIC resolves over-fitting by introducing a penalty term for the number of parameters in the model. For wave dissipation, the BIC approach identified a fit based exclusively on density as the model that best match the data while preventing over-fitting. The selection of density as the single most relevant parameter is consistent with the Sobol' indices result (Table 3).

The kinetic energy variability is also associated with shoot density changes but the percentage of explained variability (42%) is smaller than for wave dissipation. Diameter and height also contributed to changes in kinetic energy. The combination of density, height, and diameter provided the optimal fit of the data (selected by minimizing BIC) and explained 89% of variability in kinetic energy. Similar results were obtained for TKE with density explaining 45% of the variability but the combination of density, height, and diameter providing the optimal fit to TKE (selected by minimizing BIC) and explained 87% of the variability. The model response of water level variability is also best explained by a combination of density, height, and diameter (96% of water level variance explained). Thickness was not correlated with wave dissipation, kinetic energy or TKE and only contributed to water level gradient variability.

**4.4 Limitations of the current sensitivity methodology**

The model configuration chosen includes vegetation covering a small fraction of the water column to allow for proper wave dissipation. Many species of seagrass in most environments have a larger vertical footprint and can also exhibit much higher shoot densities. The goal of the study is to provide estimates on the relative importance of the different parameters through a robust sensitivity approach. The current work assumed rigid vegetation blades while the model is capable of including flexible vegetation by altering the effective blade height. The expected effect of flexible blades would be to reduce the relative importance of vegetation length ($l_v$) on the model outputs. In addition, the present work assumes a constant drag coefficient for cylindrical vegetation shape that can be a subject of the future sensitivity study. Other model parameters such as vertical and horizontal resolution, horizontal mixing parameterization, and wave and hydrodynamic forcings will also affect model results. However, our focus in this study is to analyse the sensitivity to vegetation properties and the effect of physical and numerical parameter choice goes beyond the scope of the current study.



The limitation of the Sobol' indices approach is the need to conduct a sufficient large set of model simulations to constrain the parameter space. The rate of convergence of the Sobol' estimator is slow relative to some other sensitivity analyses (Saltelli et al., 2008) but the method is chosen for its robustness and completeness. The Effective Quadratures approach allows for a reduction in the number of simulations needed to calculate the Sobol' indices, thus reducing the computational expense.

## 5 Conclusions

The coupled wave-flow-vegetation module in the COAWST modeling system provides a tool to study vegetated flows in riverine, lacustrine, estuarine and coastal environments. The resulting flow field in the presence of vegetation depends on its properties. These plant properties include - vegetation shoot density, height, diameter, and thickness. The sensitivity of the hydrodynamic and wave conditions to changes in vegetation parameters is investigated. The sensitivity analysis helps in understanding the multiparameter/multiresponse of various interactions within the model. We use a polynomial quadrature method to investigate the sensitivity of plant properties on the resulting output. The decomposition of the variance of the model solution given by the Sobol' indices is assigned to plant parameters.

The method of using Sobol' indices to quantify sensitivity can be computationally expensive. One of the goals of this work is to demonstrate a robust, practical, and efficient approach for the parameter sensitivity analysis. We show that the approach of using Effective Quadratures method to select a parameter space that is consistent with physical understanding significantly reduces the computational time required to obtain the Sobol' indices.

The evaluation of Sobol' indices shows that the input values of plant density, height and to a certain degree, diameter are consequential in determining kinetic energy, turbulent kinetic energy and water level changes. Meanwhile, the wave dissipation is mostly dependent on the variation in plant density.

The benefit of performing sensitivity analysis for vegetation model in COAWST provides guidance for future observational and modeling work to optimize their efforts without having to explore the entire parameter space. An accurate representation of processes causing kinetic energy and turbulent kinetic energy leads to enhanced understanding of sediment processes while accurate water level computations help to predict coastal flooding caused by storm surge. Similarly, wave attenuation measurements in open coasts are better understood with a correct representation of wave dissipation. In the future, we intend to perform a similar sensitivity analysis with the inclusion of biological model that will affect plant growth and the presence of vegetation would affect sediment transport. As model complexity increases with more parameters representing additional processes, input parameter sensitivity is required for the model to be applied on practical applications.

## 6 Code availability

The Effective Quadratures methodology is an open source Python based tool designed to perform sensitivity analysis for a given physical system. The instructions to install the code along with all the open source files for this tool are detailed here:





https://github.com/Effective-Quadratures/Effective-Quadratures. For any further inquiries about the Effective Quadratures methodology, please contact the corresponding author Dr. P. Seshadri (ps583@cam.ac.uk).

The COAWST model is an open source coupled hydrodynamics and wave model containing vegetation effects mainly coded in Fortran 77. This model provided the physical setting to perform the sensitivity analysis. The code is

available from https://coawstmodel-trac.sourcerepo.com after registration via email with J. C. Warner (jcwarner@usgs.gov). Any issues related to the model can be posted by going to this link: https://coawstmodel-trac.sourcerepo.com/coawstmodel_COAWST .

## 7 Data availability

The model output from various simulations used to perform sensitivity analysis in this study are available at: http://geoport.whoi.edu/thredds/catalog/clay/usgs/users/tkalra/senstivity_study/catalog.html. The link contains a "README.txt" file that explains how the folder is organized to contain model output.

## 8 Author contribution

T. S. Kalra, and N. K. Ganju designed and simulated the numerical experiment space for sensitivity analysis. P. Seshadri developed the Effective Quadratures methodology to quantify the model sensitivity. A. Beudin provided inputs on the mechanistic processes involving the vegetation model. T. S. Kalra and A. Arextabaleta performed the data analysis from the output of sensitivity study and prepared the manuscript with contributions from all co-authors.

## 9 Disclaimer

Any use of trade, firm, or product names is for descriptive purposes only and does not imply endorsement by the U.S. Government.

## 10 Acknowledgement

We thank Jeremy Testa at University of Maryland Center for Environmental Science for providing us guidance on the ranges of vegetation for sensitivity studies in the early stages of work.

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





| Process | Equation | $n_v$ | $l_v$ | $b_v$ | $t_v$ |
|---|---|---|---|---|---|
| Extraction of Momentum | $$F_{d,veg,u} = \frac{1}{2} C_D b_v n_v u \sqrt{u^2 + v^2}$$ $C_D$ = Plant drag coefficient <br> u,v = Horizontal velocity components at each vertical level | X | | X | |
| Turbulence Production <br><br> Uittenbogaard (2003) | $$P_{veg} = \sqrt{\left(F_{d,veg,u} u\right)^2 + \left(F_{d,veg,v} v\right)^2}$$ | X | | X | |
| Turbulence Dissipation <br><br> Uittenbogaard (2003) | $$D_{veg} = c_2 \frac{P_{veg}}{\tau_{eff}}$$ $$\tau_{eff} = min\,(\tau_{free}, \tau_{veg})$$ $$\tau_{free} = \frac{k}{\varepsilon}$$ $$\tau_{veg} = \left(\frac{L^2}{c_k{}^2 P_{veg}}\right)^{1/3}$$ $$L(z) = c_l \left(\frac{1 - b_v t_v n_v}{n_v}\right)^{1/2}$$ $P_{veg}$ = Turbulence Production <br> $\tau_{free}$ = Dissipation time scale of free turbulence <br> $k$ = Turbulent kinetic energy <br> $\varepsilon$ = Turbulence dissipation <br> $\tau_{veg}$ = Dissipation time scale of free turbulence <br> $$c_k = \left(c_\mu^0\right)^4 \simeq 0.09$$ $L$ = typical length scale between the plants <br> $c_l$ = Lift coefficient of order unity | X | | X | X |



| | | | | | |
|---|---|---|---|---|---|
| Wave Dissipation<br><br>Mendez and Losada (2004)<br>Dalrymple et al. (1984) | $$S_{d,veg} = \sqrt{\frac{2}{\pi}}\, g^2 \widetilde{C_D} b_v n_v \left(\frac{\tilde{k}}{\tilde{\sigma}}\right)^3$$ $$\frac{sinh^3(\tilde{k}l_v) + 3\,sinh(\tilde{k}l_v)}{3\tilde{k}cosh^3(\tilde{k}h)}\sqrt{E_{tot}}E(\sigma,\theta)$$ $\widetilde{C_D}$ = Bulk drag coefficient<br>$\tilde{k}$ = Mean wave number<br>$\tilde{\sigma}$ = Mean wave frequency<br>$h$  = Water depth<br>$E_{tot}$ = Total wave energy<br>$E$ = Wave energy at frequency $\sigma$ and direction $\theta$ | X | X | X | |
| Wave induced streaming<br><br>Luhar et al., 2010 | $$F_{s,veg} = \frac{S_{d,veg,tot}\,\tilde{k}}{\rho_0\,\tilde{\sigma}}$$ $S_{d,veg,tot}$ = Total wave energy dissipation<br>$\tilde{k}$ = Mean wave number<br>$\tilde{\sigma}$ = Mean wave frequency<br>$\rho_0$ = Reference density of seawater | X | X | X | |

**Table 1: Processes in ROMS and SWAN to model the presence of vegetation. The different input parameters (density, $n_v$; height, $l_v$; diameter, $b_v$; and thickness, $t_v$) affecting model wave and hydrodynamics are included.**

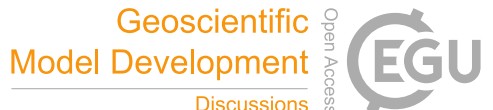

| | Density (stems/m$^2$) $n_v$ | Height (m) $l_v$ | Diameter (mm) $b_v$ | Thickness (mm) $t_v$ |
|---|---|---|---|---|
| 1. | 144.3 | 0.24 | 6.0 | 0.6 |
| 2. | 62.1 | 0.17 | 3.0 | 0.6 |
| 3. | 226.5 | 0.3 | 6.0 | 0.3 |
| 4. | 144.3 | 0.17 | 9.0 | 0.9 |
| 5. | 144.3 | 0.3 | 3.0 | 0.9 |
| 6. | 62.1 | 0.3 | 6.0 | 0.3 |
| 7. | 226.5 | 0.17 | 6.0 | 0.3 |
| 8. | 144.3 | 0.3 | 9.0 | 0.9 |
| 9. | 144.3 | 0.24 | 9.0 | 0.3 |
| 10. | 226.5 | 0.24 | 6.0 | 0.9 |
| 11. | 144.3 | 0.24 | 3.0 | 0.3 |
| 12. | 62.1 | 0.24 | 6.0 | 0.9 |
| 13. | 62.1 | 0.17 | 6.0 | 0.3 |
| 14. | 62.1 | 0.24 | 9.0 | 0.6 |
| 15. | 144.3 | 0.17 | 3.0 | 0.9 |

**Table 2: Plant property input combination for different simulations during sensitivity analysis.**





|  | Plant Density $n_v$ | Plant Height $l_v$ | Plant Diameter $b_v$ | Plant Thickness $t_v$ |
|---|---|---|---|---|
| Wave dissipation | 0.68 | 0.24 | 0.032 | 0.01 |
| Kinetic energy | 0.36 | 0.44 | 0.12 | 0.03 |
| Maximum water level change | 0.38 | 0.43 | 0.15 | 0.01 |
| Turbulent kinetic energy | 0.35 | 0.42 | 0.12 | 0.03 |

**Table 3: Sobol Indices for all the outputs.**

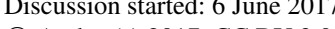

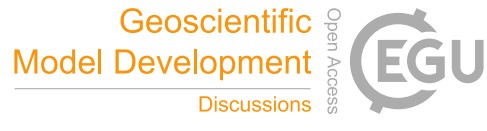

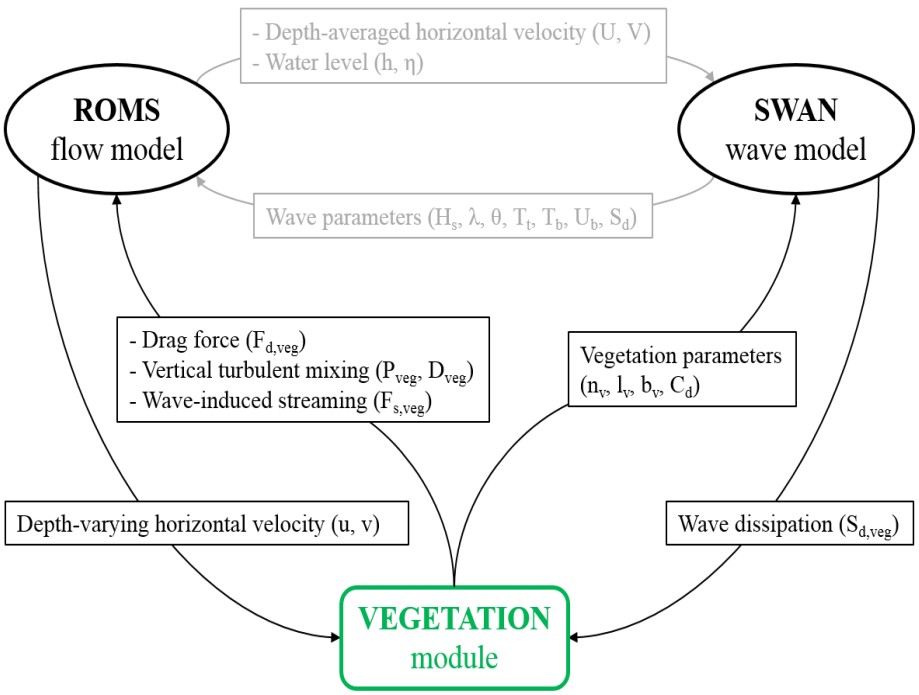

**Figure 1: Schematic showing the vegetation module implementation in COAWST model (Figure adapted from Beudin et. al, 2016)**



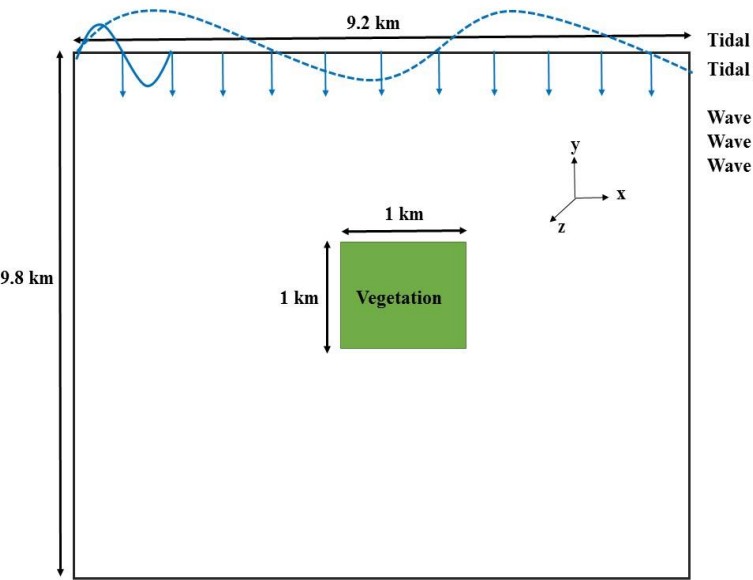




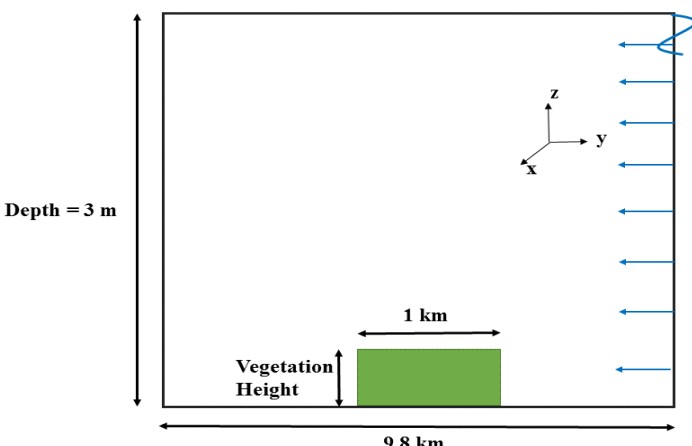

**Figure 2: Schematic showing the idealized domain (Not drawn to scale)**

**(a) Plan view and (b) Cross-sectional view.**





**Figure 3: Percentage change from minimum for the four impact parameters: a) wave dissipation; b) kinetic energy; c) maximum water level change (WL); and d) Turbulent Kinetic Energy (TKE).**



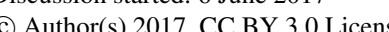

**Figure 4: Standard deviation from wave dissipation (W m$^{-2}$) in presence of vegetation (Plan view). The area of the vegetation patch is highlighted in the middle of the domain.**





5    **Figure 5: Standard deviation in kinetic energy (cm² s⁻¹) in presence of vegetation (cross-sectional view).**





**Figure 6: Standard deviation in water level in presence of vegetation (plan view). The area of the vegetation patch is highlighted in the middle of the domain.**



**Figure 7: Standard deviation in Turbulent Kinetic Energy (TKE) (cm² s⁻¹) in presence of vegetation (plan view). The area of the vegetation patch is highlighted in the middle of the domain.**



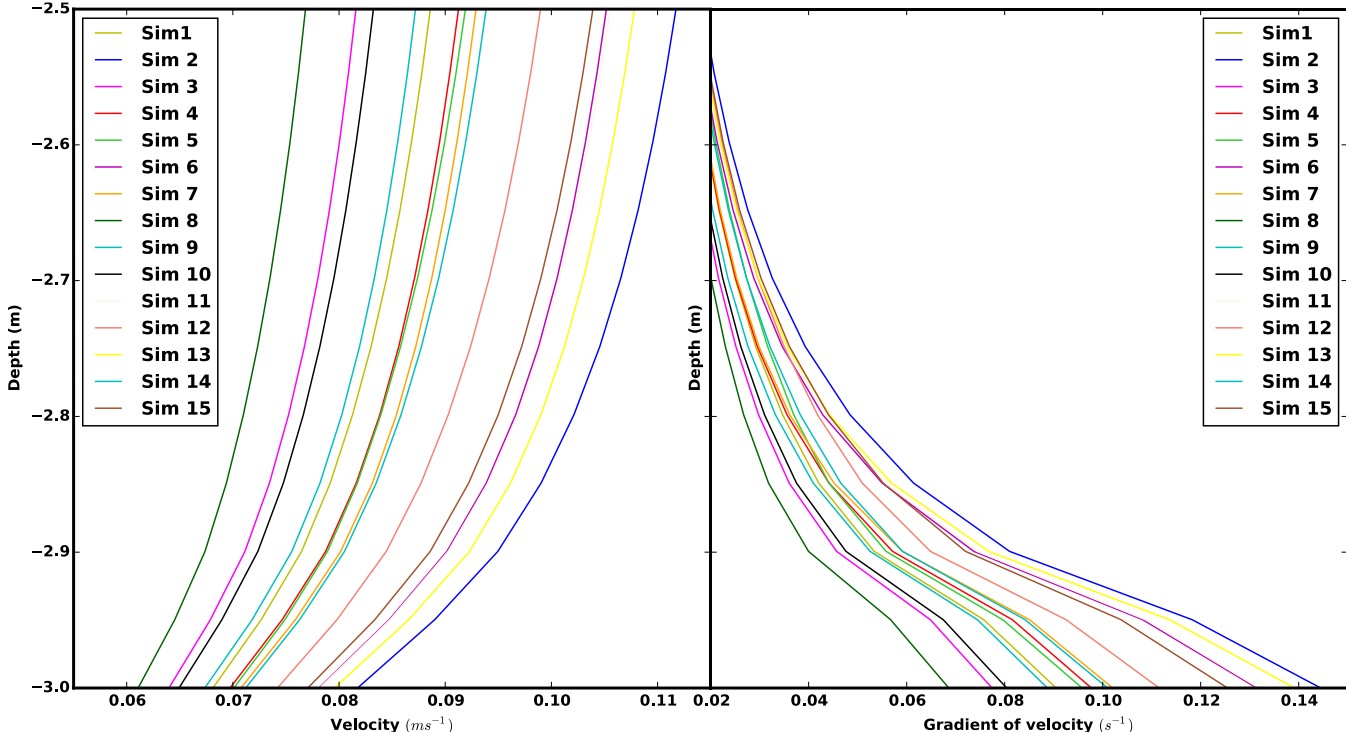

**Figure 8: (a) Velocity (m s$^{-1}$) profile and (b) Vertical gradient (s$^{-1}$) of velocity profile varying with depth in front of the vegetation patch at a particular time instance during flood for different simulations.**

