# Peer review of "Sensitivity Analysis of a Coupled Hydrodynamic-Vegetation Model Using the Effectively Subsampled Quadratures Method (ESQM v5.2)"

_Geoscientific Model Development, 2017_

## Short Comment (SC1) · 8 Jun 2017

Dear authors,

In my role as Executive editor of GMD, I would like to bring to your attention our Editorial version 1.1:

http://www.geosci-model-dev.net/8/3487/2015/gmd-8-3487-2015.html

This highlights some requirements of papers published in GMD, which is also available on the GMD website in the 'Manuscript Types' section:

http://www.geoscientific-model-development.net/submission/manuscript_types.html

[Figure]

In particular, please note that for your paper, the following requirements have not been met in the Discussions paper:

- "The main paper must give the model name and version number (or other unique identifier) in the title."

- "If the model development relates to a single model then the model name and the version number must be included in the title of the paper. If the main intention of an article is to make a general (i.e. model independent) statement about the usefulness of a new development, but the usefulness is shown with the help of one specific model, the model name and version number must be stated in the title. The title could have a form such as, "Title outlining amazing generic advance: a case study with Model XXX (version Y)"."

- "All papers must include a section, at the end of the paper, entitled 'Code availability'. Here, either instructions for obtaining the code, or the reasons why the code is not available should be clearly stated. It is preferred for the code to be uploaded as a supplement or to be made available at a data repository with an associated DOI (digital object identifier) for the exact model version described in the paper. Alternatively, for established models, there may be an existing means of accessing the code through a particular system. In this case, there must exist a means of permanently accessing the precise model version described in the paper. In some cases, authors may prefer to put models on their own website, or to act as a point of contact for obtaining the code. Given the impermanence of websites and email addresses, this is not encouraged, and authors should consider improving the availability with a more permanent arrangement. After the paper is accepted the model archive should be updated to include a link to the GMD paper."

- Descriptions of software tools are subject to the same criteria as model descriptions

Therefore, please name (and number) the model you do the assessment for in the title of the manuscript in the revised submission to GMD. If there is an acroynm for the "Effectively Subsampled Quadratures Method" and a version number of the tool containing it, they should also be named in the title.

Additionally, please consider publishing the method tools in a data repository (i.e. with an associated DOI)

Yours,

Astrid Kerkweg

───────────────────────

---

## Author Comment (AC1) · 9 Jun 2017

Dear Executive Editor,

Thank you for your comments on the discussion paper. As per your suggestions:

The title of the discussion paper can be modified to include the name of the model as an acronym : Sensitivity Analysis of Vegetation Module in the Coupled Ocean Atmosphere Waves Sediment Transport (COAWST) Model Using the Effectively Subsampled Quadratures Method

Some parts of the COAWST model are still under development and we are in the

process of getting a DOI for it in the coming months. We utilize the version number of the model only during the distribution process through "Subversion client" and the version number associated with the distribution that we used for this analysis is "svn 1108". We can add this information in the "Code Availability section".

In addition, we can add the information for the methods that we used in the "Code availability section" along with its DOI specification as:

Effective Quadratures Version 5.2 (https://github.com/Effective-Quadratures/Effective-Quadratures/releases). Seshadri, P., Parks, G., (2017) "Effective Quadratures (EQ): Polynomials for Computational Engineering Studies", Journal of Open Source Software, 2(11). DOI: 10.21105/joss.00166

Since our discussion paper is already downloadable, we request you to add a disclaimer section in the content of the paper while it finishes review as per United States Geological Survey (USGS) policy. This is the content of the disclaimer section:

This draft manuscript is distributed solely for purposes of scientific peer review. Its content is deliberative and pre-decisional, so it must not be disclosed or released by reviewers. Because the manuscript has not yet been approved for publication by the U.S. Geological Survey (USGS), it does not represent any official USGS finding or policy.

Thank you for your assistance.

———————————————

---

## Short Comment (SC2) · 19 Jun 2017

Dear Authors,

this is the wrong place for your request for a disclaimer.

*"Since our discussion paper is already downloadable, we request you to add a disclaimer section in the content of the paper while it finishes review as per United States Geological Survey (USGS) policy. This is the content of the disclaimer section: This draft manuscript is distributed solely for purposes of scientific peer review. Its content is deliberative and pre-decisional, so it must not be disclosed or released by review-*

*ers. Because the manuscript has not yet been approved for publication by the U.S. Geological Survey (USGS), it does not represent any official USGS finding or policy."*

First, I wonder, why you did not add this disclaimer yourselves, as you should have known, that you publish your article in a journal with public discussion before final release.
Second, changes to manuscripts can only be achieved by the publication office, so please ask them to add the disclaimer. However, usually manuscripts are not changed after they are online.

Best reagrds, Astrid Kerkweg

―――――――――――――――――――――

---

## Editor Comment (EC1) · P.R. Halloran (Editor) · 19 Jun 2017

Dear Authors and Executive Editor,

Many thanks to Astrid Kerkweg for highlighting these points and following up on the disclaimer issue. I write to follow up on the comment about adding the model version to the title.

Your manuscript was submitted as an 'Methods for assessment of models', which I feel accurately describes the manuscript. As you will see here:

http://www.geoscientific-model-development.net/about/manuscript_types.html

the requirement 'The main paper must give the model name and version number (or other unique identifier) in the title' only applies to model description papers. It is considered that this title structure helps the process of documenting model development clearly. Thinking about your manuscript, the reference to 'COAWST' is not documenting model development undertaken in your manuscript, so I would not worry about adding this. If you feel that the analysis tool you have developed are likely to evolve and be used directly by others (rather than them producing their own versions of these tools), perhaps I could encourage you to consider a title along the lines of 'Sensitivity Analysis of... Using the Effectively Subsampled Quadratures Method (ESQM vx.y)'?

Many thanks, Paul Halloran (Editor)

―――――――――――――――――

---

## Author Comment (AC2) · 20 Jun 2017

Dear Editors,

We concur by the comments made by the editor. We can include the version number of the analysis tool whilst not including "COAWST" in the title for the reasons that the editor rightly pointed out.

The title can be mentioned as: Sensitivity Analysis of a Coupled Hydrodynamic-Vegetation Model Using the Effectively Subsampled Quadratures Method (ESQM v5.2)

Once again, we appreciate your inputs.

[Figure]

Thanks, Tarandeep
* * *

---

## Referee Comment (RC1) · Anonymous Referee #1 · 12 Jul 2017

Sensitivity Analysis of a Coupled Hydrodynamic-Vegetation Model Using the Effectively Subsampled Quadratures Method

Kalra et al

This paper details a sensitivity analysis of aquatic vegetation in the COAWST model using a novel Effective Quadratures method. The model uses a three-dimensional drag term and generates TKE in the presence of vegetation. The paper does not detail the implementation details of the vegetation module (though equations are given in Table 1), but evaluates the sensitivity of the model via novel Effective Quadratures method. The outcome of this paper is guidance on setting parameters in similar vegetation

modules.

Major corrections ——————

The paper is not focused in its current state. The details of the EQ methods are not given and neither are details of the vegetation model. In order to properly judge the conclusions more details on the EQ methodology are required. The paper therefore lacks a clear aim: is it detailing the EQ method (no - this is cited as Seshadri et al, 2017b, although as a paper in JOSS it lacks detail), the coupling of vegetation to COAWST (no - this is Beudin et al 2017). The paper should therefore be refocused along the lines of: "new methods for assessment of models, including work on developing new metrics for assessing model performance and novel ways of comparing model results with observational data" as to my knowledge EQ has not been used in a coastal model and as such this would represent an advance. More details of the implementation would greatly improve the paper.

Minor corrections ——————

Title: Change as requested by Editor

Line 4 - extra ) after parameters

Line 15 40 CPU hours. Is that 40 hours * 24 cores? Or 40/24 hours? Not clear.

Figures 4 to 7: Colour scheme is not suitable for colour-blind readers and also has the potential to produce artificial "highlights" due to the luminosity changes. For a continuous scales as used in all plots a continuous colour scheme should be used. See https://matplotlib.org/users/colormaps.html and https://bids.github.io/colormap/ for examples

Figure 8: Difficult to differentiate the lines, especially those with pale colours (e.g. sim 13). Can the lines have a label placed on them (or nearby) to aid the reader?

Code availability: https://coawstmodel-trac.sourcerepo.com/ gave an In

---

## Referee Comment (RC2) · Anonymous Referee #2 · 23 Aug 2017

General comments Indeed, this type of models are needed for coastal studies. And the general idea of having more efficient sensitivity analysis methods for this type of models is very attractive. But: the real benefits are not imminent from this MS, as the real details of both model and method are not described (i.e. not informative) - it is too easy to lay this MS aside as just another paper describing some sort of method. Remember that the people interested in using (the results of) this type of models are not necessarily interested in complicated sensitivity studies; what can you do to make their life easier?

COAWST contains the word sediment, but the effects of vegetation on sediment are

not mentioned in the abstract (or dealt with in the MS?)

The abstract describes the results very vaguely without any causality, and does not say anything we did not expect already.

For the demonstration of this technique, seagrass properties -which can be measured quite well, i.e. do not have large uncertainty- have been varied over a relatively small range, whereas the environmental conditions have not been varied. Rather than learning which details matter, it would be interesting to see when (under which conditions) these details matter; try to compare the combinations of veg parameters to literature on flow regimes, e.g. Mitul & Nepf 2013. My gut feeling says more uncertain parameters like $C_d$ (=1 which is ok for a rigid cylinder, not for flexible, flat-bladed seagras!!) and $z_0$ can have stronger effects. Such considerations are mentioned in section 4.4, but should be discussed earlier to avoid loosing your public.

Why use pct change from the minimum value? That is a rather extreme situation.

Specific comments p2_l20 no drag coefficient or spatial density? Note that in the SWAN implementation (Suzuki et al), some parameters have exactly the same effect in the energy dissipation equations, see http://swanmodel.sourceforge.net/online_doc/swantech/node21.html. p3_l1-15 what is the overall message of these loose examples? p4 why not refer to Table 1 for the equations? p4_l29 , instead of ; p5_l28-31 is this the stem density or the leaf density? Typically, Zostera marina has multiple leaves, and as the stem is usually short it may be the leaves that interact with the flow. For leaf density, this is a very low number but it matches the diameter (=leaf width?). The thickness of the leaves is rather large in my opinion, given the small length of these plants. How has this been measured? With a caliper or estimated? (personal comm is not published data!)

p18, Table 2 What about ah (as in Nepf, 2012)? p20, Fig 1 Are the Drag force, mixing and streaming calculated by the vegetation module? I would think these are hydrodynamic properties computed by ROMS, based on the same set of veg parameters that

go to SWAN. p28, in caption: where can I find the condtions for these sims? I am surprised the classical S shape for flow in/over canopies is missing.

---

## Author Response (AR1)

Dear Editor,

5  On behalf of my coauthors, I am pleased to resubmit the manuscript "Sensitivity Analysis of a Coupled Hydrodynamic-Vegetation Model Using the Effectively Subsampled Quadratures Method (ESQM v5.2)" to be considered for publication in Geophysical Model Development.

The manuscript provides an evaluation of a model to simulate 3D effects of submerged aquatic vegetation within
10  a coupled hydrodynamics and wave model. Using a novel method to perform sensitivity analysis, we analyse the sensitivity of the vegetation model to provide guidance to optimizing efforts and reducing parameter space for future observational and modeling work. We think that the manuscript has the potential to be useful for the coastal modeling community in general and more specifically scientists trying to provide informed advice to coastal managers about the benefits of green infrastructure in reducing the impacts of large storms in coastal
15  environments.

Following the publishing process at USGS, the discussion paper was internally reviewed by a USGS scientist. After the suggestions of the internal reviewer and external reviewers were incorporated, it was reviewed to ensure that the standards of USGS policy of peer review were met.

The issues raised with the previous submission have been addressed following the Reviewers' guidance. The reviewers major concern was to address the issue of describing the vegetation model and the sensitivity method. Along with the clarification that these models were not developed as a part of this study and were only being used, we have added additional text to clarify this point to the readers. We have incorporated all the other
25  changes based on the reviewers' suggestions and addressed their comments in details. Please find the documents that address the suggestions and comments from all the referees. The sub-sections in the document contain the following:

         1.1 Comments from referee 1, Page 2
30       1.2 Comments from referee 2, Page 3
         2.1 Authors' point by point response from referee 1, Page 4-9
         2.2 Authors' point by point response from referee 2, Page 9-15
         3.1 Authors' changes in the manuscript with markups, Page 16-49
         4.1 Final manuscript with all the changes included, Page 50-76

         In addition, we want to specifically mention that a comment by the anonymous reviewer 2 was made to add a reference that was used in selecting the range of "vegetation thickness". We had responded to reviewer 2 during the interactive discussion that that we had used personal communication from Jeremy Testa, who is an assistant professor at the University of Maryland's Center for Environmental Science. However, the USGS
40  Bureau pointed out that personal communication was no longer allowed in USGS publications. Therefore, we have replaced the reference of personal communication to the citing of a paper that contains the relevant information on the range of "thickness".

Sincerely,

Tarandeep

**1.1 Comments from Referee 1**

This paper details a sensitivity analysis of aquatic vegetation in the COAWST model using a novel
Effective Quadratures method. The model uses a three-dimensional drag term and generates TKE in
15 presence of vegetation. The paper does not detail the implementation details of the vegetation module
(though equations are given in Table 1), but evaluates the sensitivity of the model via novel Effective
Quadratures method.
The outcome of this paper is guidance on setting parameters in similar vegetation modules.

20 Major corrections ————————

The paper is not focused in its current state. The details of the EQ methods are not given and neither are
details of the vegetation model. In order to properly judge the conclusions more details on the EQ
methodology are required. The paper therefore lacks a clear aim: is it detailing the EQ method (no - this
25 is cited as Seshadri et al, 2017b, although as a paper in JOSS it lacks detail), the coupling of vegetation
to
COAWST (no - this is Beudin et al 2017). The paper should therefore be refocused along the lines of:
"new methods for assessment of models, including work on developing new metrics for assessing
model performance and novel ways of comparing model results with observational data" as to my
30 knowledge EQ has not been used in a coastal model and as such this would represent an advance. More
details of the implementation would greatly improve the paper.

Minor corrections ————————

35 Title: Change as requested by Editor

Line 4 - extra ) after parameters

Line 15 40 CPU hours. Is that 40 hours * 24 cores? Or 40/24 hours? Not clear.

Figures 4 to 7: Colour scheme is not suitable for colour-blind readers and also has the potential to
produce artificial "highlights" due to the luminosity changes. For a continuous scales as used in all plots

a continuous colour scheme should be used. See https://matplotlib.org/users/colormaps.html and https://bids.github.io/colormap/ for
examples

5  Figure 8: Difficult to differentiate the lines, especially those with pale colours (e.g. sim 13).

Can the lines have a label placed on them (or nearby) to aid the reader? Code availability: https://coawstmodel-trac.sourcerepo.com/ gave an error

**1.2 Comments from Referee 2**

15  General comments Indeed, this type of models are needed for coastal studies. And the general idea of having more efficient sensitivity analysis methods for this type of models is very attractive. But: the real benefits are not imminent from this MS, as the real details of both model and method are not described (i.e. not informative) - it is too easy to lay this MS aside as just another paper describing some sort of method.

20  Remember that the people interested in using (the results of) this type of models are not necessarily interested in complicated sensitivity studies; what can you do to make their life easier?
COAWST contains the word sediment, but the effects of vegetation on sediment are not mentioned in

the abstract (or dealt with in the MS?)

The abstract describes the results very vaguely without any causality, and does not say anything we did
25  not expect already.

For the demonstration of this technique, seagrass properties -which can be measured quite well, i.e. do not have large uncertainty- have been varied over a relatively small range, whereas the environmental conditions have not been varied. Rather than learning which details matter, it would be interesting to see
30  when (under which conditions) these details matter; try to compare the combinations of veg parameters to literature on flow regimes, e.g. Mitul & Nepf 2013. My gut feeling says more uncertain parameters like $C_d$ (=1 which is ok for a rigid cylinder, not for flexible, flat-bladed seagras!!) and $z_0$ can have stronger effects. Such considerations are mentioned in section 4.4, but should be discussed earlier to avoid loosing your public.
35
Why use pct change from the minimum value? That is a rather extreme situation.

Specific comments p2_l20 no drag coefficient or spatial density? Note that in the SWAN implementation (Suzuki et al), some parameters have exactly the same effect in the energy dissipation equations, see
http://swanmodel.sourceforge.net/online_doc/swantech/node21.html.

p3_l1-15 what is the overall message of these loose examples? p4 why not refer to Table 1 for the equations?

p4_l29 , instead of ; p5_l28-31 is this the stem density or the leaf density? Typically, Zostera marina has multiple leaves, and as the stem is usually short it may be the leaves that interact with the flow. For leaf density, this is a very low number but it matches the diameter (=leaf width?). The thickness of the leaves is rather large in my opinion, given the small length of these plants. How has this been measured? With a caliper or estimated? (personal comm is not published data!)

p18, Table 2 What about ah (as in Nepf, 2012)? p20, Fig 1 Are the Drag force, mixing and streaming calculated by the vegetation module? I would think these are hydrodynamic properties computed by ROMS, based on the same set of veg parameters that go to SWAN. p28, in caption: where can I find the conditions for these sims? I am surprised the classical S shape for flow in/over canopies is missing.

**2.1 Author response to Referee 1**

The response to the Reviewer's comments are in black while the original comments are in blue.

This paper details a sensitivity analysis of aquatic vegetation in the COAWST model using a novel Effective Quadratures method. The model uses a three-dimensional drag term and generates TKE in the presence of vegetation. The paper does not detail the implementation details of the vegetation module (though equations are given in Table 1), but evaluates the sensitivity of the model via novel Effective Quadratures method. The outcome of this paper is guidance on setting parameters in similar modules.

Major corrections ——————
The paper is not focused in its current state. The details of the EQ methods are not given and neither are details of the vegetation model. In order to properly judge the conclusions more details on the EQ methodology are required. The paper therefore lacks a clear aim: is it detailing the EQ method (no - this is cited as Seshadri et al, 2017b, although as a paper in JOSS it lacks detail), the coupling of vegetation to COAWST (no - this is Beudin et al 2017). The paper should therefore be refocused along the lines of: "new methods for assessment of models, including work on developing new metrics for assessing model performance and novel ways of comparing model results with observational data" as to my knowledge EQ has not been used in a coastal model and as such this would represent an advance. More details of the implementation would greatly improve the paper.

The paper does not detail the implementation details of the vegetation module because a reference paper also published by the part of the same group of authors (Beudin et al., 2017) contains the details of the implementation of the vegetation model. We have attempted to summarize the physical processes and related equations in Table 1 of the paper. The goal of the current paper is to provide guidance to the setting up of the parameters for the vegetation model by performing the sensitivity analysis.

The Reviewer states that the JOSS paper (Seshadri et al., 2017b) lacks details of the EQ method. This is true, however, the JOSS paper is simply for the software release. The underlying methodology is cited in the SIAM paper (Seshadri et al., 2017a) in Section 2.2, where more details as to what the subsampling strategy entails are provided. The computation of the Sobol' indices is outlined in Surdet et al., but is also provided in the appendix of the SIAM paper (Seshadri et al., 2017a). The purpose of the present paper is to demonstrate the computation of sensitivity metrics using an approach that is amenable to multi-physics simulations that are not computationally cheap. The EQ method has not been developed as a part of this study and we are not comparing the model results with observational data as well.

In the final version of the manuscript we will modify the introduction and conclusions to clarify the focus following the Reviewer's suggestions. Specific additions to the text include:

Line 25-29 Page 3 in introduction: These tools are implemented in the open source package, Effective Quadratures method (EQ) (Seshadri et al., 2017b) and the current work provides one of the first applications of this methodology to quantify sensitivity of input parameters in coastal models.
Therefore, the goal of this work is to use the EQ method to estimate Sobol' indices to estimate the sensitivity of the flow and wave dynamics to vegetation parameters in COAWST model.
Line 32 Page 10 in conclusions: We use an existing tool comprising of polynomial quadrature method to investigate the sensitivity of plant input properties for the vegetation module in COAWST model.

Minor Corrections:
  1. Title: Change as requested by Editor
     Changed the title of the paper upon Editor's suggestion to include the version of EQ method

"Sensitivity Analysis of a Coupled Hydrodynamic-Vegetation Model Using the Effectively Subsampled Quadratures Method (ESQM v5.2) "

2. Line 4 fixed the typo

3. Line 15, Yes it is 40 CPU hours for 24 cores. Alternatively, we can use the word "Core hours". So the current simulations required 24*40 core hours.

4. Color scheme in Figures 4-7 is now using a continuous color scheme "viridis" from the matplotlib tool kit options. As an example, Figure 4 is shown on Page 3 of this document:

5. Figure now contains the individual legend on the line. Also, replaced the lines with different colors with symbols and markers for clarity. Please see the new figure at the end of this response document.

6. Code availability link has been modified to contain the right information.
https://coawstmodel-trac.sourcerepo.com/coawstmodel_COAWST

[Figure]

**Figure 4: Standard deviation from wave dissipation (W m⁻²) in presence of vegetation (Plan view). The area of the vegetation patch is highlighted in the middle of the domain.**

[Figure]

**Figure 8: (a) Velocity (m s⁻¹) profile and (b) Vertical gradient (s⁻¹) of velocity profile varying with depth in front of the vegetation patch at a particular time instance during flood for different simulations.**

**2.2 Author response to Referee 2**

5 The response to the Reviewer's comments are in black while the original comments are in blue.

*General comments Indeed, this type of models are needed for coastal studies. And the general idea of having more efficient sensitivity analysis methods for this type of models is very attractive. But: the real benefits are not imminent from this MS, as the real details of both model and method are not described (i.e. not informative) - it is too easy to lay this MS aside as just another paper describing some sort of*
10 *method. Remember that the people interested in using (the results of) this type of models are not necessarily interested in complicated sensitivity studies; what can you do to make their life easier?*

Response 1: The goal of the current paper is to provide guidance to the setting up of the parameters for the vegetation model by performing sensitivity analysis. The paper does not detail the implementation details of the vegetation module because a referenced paper also published by the same group of authors
15 (Beudin et al., 2017) contains the details of the implementation of the vegetation model. We have attempted to summarize the physical processes and related equations in Table 1 of the paper. While the Beudin et al. paper was able to describe the details of the vegetation module, as the Reviewer mentions, users need some guidance to "make their life easier" when choosing the parameters of a model. The present MS provides that guidance and evaluates the repercussions of the parameter choice through
20 sensitivity analysis.

The chosen methodology of sensitivity analysis is detailed and cited in the Seshadri et al. 2017a paper of one of the co-authors, where more details as to what the subsampling strategy entails are provided. The computation of the Sobol' indices is outlined in Surdet et al., but is also provided in the appendix of the SIAM paper (Seshadri et al., 2017a). The purpose of the present paper is to demonstrate
25 the computation of sensitivity metrics using an approach that is amenable to multi-physics simulations that are not computationally cheap.

In the new version of the manuscript we have modified the introduction and conclusions to clarify the idea of the paper. Specific additions to the text include:

"Page 3 Line 25-29 in introduction: "These tools are implemented in the open source package, Effective Quadratures Method (EQ) (Seshadri et al., 2017b) and the current work provides one of the first applications of this methodology to quantify sensitivity of input parameters in coastal models.

Therefore, the goal of the present work is to take advantage of the EQ method to provide Sobol' indices that quantify the sensitivity of the flow and wave dynamics to vegetation parameters in COAWST model."

Page 10 Line 32 in conclusions: "We use a recently developed tool that formulates the Effective Quadratures method to quantify the sensitivity of plant input properties for the vegetation module in COAWST."

*COAWST contains the word sediment, but the effects of vegetation on sediment are not mentioned in the abstract (or dealt with in the MS?)*

Response 2: It is true that COAWST can model the transport of sediment. It is also able to model atmospheric conditions (WRF), but both sediment and atmospheric processes go beyond the scope of the present paper. The chosen output parameters for sensitivity analysis such as kinetic energy highlight the first order effects of introducing vegetation in the flow domain. While the model is capable of coupling the effects of vegetation with sediment model, the idea of the current study is not to model physical mechanisms due to the presence of vegetation. The focus of the current study remains to demonstrate the influence of vegetation parameter choice on the flow field. The resulting sediment transport effects are left for future work.

*For the demonstration of this technique, seagrass properties -which can be measured quite well, i.e. do not have large uncertainty- have been varied over a relatively small range, whereas the environmental conditions have not been varied. Rather than learning which details matter, it would be interesting to see when (under which conditions) these details matter; try to compare the combinations of veg parameters to literature on flow regimes, e.g. Mitul & Nepf 2013.*

Response 3: We believe the present work is relevant as it characterizes the sensitivity to seagrass properties because they do entail large uncertainty for measurement purposes and also exhibit a large range of spatial and temporal variability. Adding to the uncertainty, the seagrass growth is dynamic depending on nutrient supply, light quality and availability, and algal presence.

In Mitul & Nepf (2013), the focus is on creating situations ranging from blade scale to reach scale to understand the changes in resulting flow field. This kind of study is relevant for theoretical understanding of vegetation effects. The purpose of the vegetation module in COAWST is to be able to include the vegetation effects in realistic model applications for areas with known seagrass characteristics. The future goal is to use the model for the study of realistic changes in water level and wave attenuation triggered by the presence of vegetation.

*My gut feeling says more uncertain parameters like C_d (=1 which is ok for a rigid cylinder, not for flexible, flat-bladed seagras!!) and z_0 can have stronger effects. Such considerations are mentioned in section 4.4, but should be discussed earlier to avoid loosing your public.*

Response 4: We agree with the Reviewer that the sensitivity of drag coefficient can be studied. The determination of drag coefficient is usually done in controlled environments (laboratory scale). However, the purpose of the current work is to provide guidance for the vegetation parameters that are measured in physical systems and are required in the COAWST model for modeling the effect of seagrass. Variations on C_d are unlikely to be measured in the field and thus users could rely on the published literature for an appropriate choice based on the type, shape, and flexibility of the vegetation under study.

Following the reviewer's comment, we have added the reasons for not considering the input of C_d in the sensitivity of vegetation model in an earlier section on Page 6, Lines 2-6 (Section 2.3) as: "In addition to these four vegetation properties, the vegetative model requires an input of drag coefficient $(C_D)$ in the flow model and the wave model. However, variations on $C_D$ are unlikely to be measured in the field and thus users could rely on the published literature for an appropriate choice based on the type, shape, and flexibility of the vegetation under study."

*Why use pct change from the minimum value? That is a rather extreme situation.*

Response 5: The thoughts behind using a minimum value is to show a relative change from each of the simulations with different vegetation parameters. Also, the mean of the simulations might not be as relevant, if response is non-linear.

*Specific comments p2_l20 no drag coefficient or spatial density? Note that in the SWAN implementation (Suzuki et al), som parameters have exactly the same effect in the energy dissipation equations, see http://swanmodel.sourceforge.net/online_doc/swantech/node21.html.*

10   Response 6: The bulk drag coefficient required in the SWAN parameterization in COAWST model assumes a constant drag coefficient. The spatial stem density is variable. Following the reviewer's specific comments Page 4 line 16-17 include the following text: "The parameterization of SWAN to account for wave dissipation implemented by Suzuki et al. 2012 has the same effect as energy dissipation."

*p3_l1-15 what is the overall message of these loose examples?*

Response 7: The overall message of providing previous literature on sensitivity studies involving physical systems is to lead to the idea that the choice of sensitivity analysis method depends on multiple factors such as computational costs, characteristics of the model, number of input parameters, and/or
20   potential interactions between parameters. The authors have rephrased the paragraph to make the context of providing the examples in the new version on Page 3 Line 12 to 18 in Introduction as:

"One of the challenges associated with a Monte Carlo approach to computing the Sobol' indices is the large number of model evaluations required for approximating conditional variance.

All these studies highlight various approaches to perform sensitivity analysis. Saltelli et al.
25   (2008) provided a comparison of different sensitivity analysis methodologies and the optimal setup for specific combinations of parameters and model.   Ultimately, the choice of sensitivity analysis methodology depends on multiple factors such as the computational cost of running the model, the

characteristics of the model (e.g., nonlinearity), the number of input parameters, and/or the potential interactions between parameters."

*p4 why not refer to Table 1 for the equations? p4_l29 , instead of ;*

Response 8: We have added the reference to table 1 on page 4 in line 18 as:

"The parameterizations used to implement the effect of vegetation in both ROMS and SWAN models are mentioned in Table 1 and detailed in Beudin et al. (2017)."

*p5_l28-31 is this the stem density or the leaf density? Typically, Zostera marina has multiple leaves, and as the stem is usually short it may be the leaves that interact with the flow. For leaf density, this is a very low number but it matches the diameter (=leaf width?).*

Response 9: It is the stem density. Hence the units are stems/m$^2$. We have replaced the word "density" by "stem density" in the paper for clarity. The leaf width = diameter.

*The thickness of the leaves is rather large in my opinion, given the small length of these plants. How has this been measured? With a caliper or estimated? (personal comm is not published data!*

Response 10: This has been an estimation. Considering that the chosen range is large, even then the sensitivity analysis shows that thickness is not causing any changes to the flow field (i.e. the chosen model output parameters are the least sensitive to thickness).

Jeremy Testa is a Assistant Professor in the Chesapeake Biological Laboratory at University of Maryland Center for Environmental Science. Some of the relevant literature of Jeremy Testa includes:

1. "Temporal responses of coastal hypoxia to nutrient loading and physical controls", Biogeosciences 6 (12), 2985-3008, 2009.
2. "The metabolism of aquatic ecosystems: history, applications, and future challenges", Aquatic Sciences 74 (1), 15-29, 2012.
3. "Submersed aquatic vegetation in Chesapeake Bay: sentinel species in a changing world", BioScience, Volume 67, Issue 8, 1 August 2017, Pages 698–712, 2017.

*p18, Table 2 What about ah (as in Nepf, 2012)?*

Response 11: In Nepf, 2012; "ah" helps in distinguishing the different flow regime where "a" is the frontal area per unit canopy volume. The canopy volume is based on the spacing between elements. The

implementation of vegetation model does not work on the scales to resolve the spacing between the elements and only considers vegetation height "h" in the equations for modeling seagrass.

*p20, Fig 1 Are the Drag force, mixing and streaming calculated by the vegetation module? I would think these are hydrodynamic properties computed by ROMS, based on the same set of veg eparameters that go to SWAN.*

Response 12: This is absolutely correct. The equations of drag, mixing and streaming computed in ROMS use the same vegetation parameters as SWAN. There are two parts of the vegetation module, one in SWAN and one in ROMS and the two exchange information in the coupled system, thus resolving the wave-current modifications.

*p28, in caption: where can I find the conditions for these sims?*

Response 13: The different simulations are characterized by varying the vegetation properties (mentioned in Table 2.) while keeping the baseline setup similar in all cases for the purpose of studying their sensitivity. The discussion on the choice of the vegetation properties is detailed in Section 2.3. The domain setup, initial, and boundary conditions are the same for all simulations (except for the variation of vegetation properties). This setup is described in details in Section 2.4 (and schematically described in figure 2).

*I am surprised the classical S shape for flow in/over canopies is missing*

Response 14: The reason we are missing a S shape in the flow in/over the canopy is because the chosen vegetation stem density range is not large enough to cause that shape. The original paper showing the vegetation model implementation by the current authors (Beudin et al., 2017) shows this shape when a vegetation stem density of 2500 stems/m$^2$ is chosen (Figure 4a).

[Figure]

Figure adapted from Beudin et al. (2017) showing vertical profiles of a) mean flow velocity, and b) turbulent Reynolds shear stress $(\overline{v'w'}=-KM\delta v\delta z)$ in the middle of the patch at peak flood without vegetation (solid black), with stiff vegetation (solid blue), and with flexible vegetation (dash green).

**3.1 Marked up manuscript that tracks changes in MS-Word Format including the changes:**

[revised manuscript text omitted]

**Commented [KTS3]:** Following the reviewer 2's suggestion, have added the reasons for not considering the input of C_d in t sensitivity of vegetation model.

[revised manuscript text omitted]